

# Revisited heat budget and probability distributions of turbulent heat fluxes in the Mediterranean Sea

Mahmud Hasan Ghani[1], Nadia Pinardi[1,2], Antonio Navarra[2,3], Lorenzo Mentaschi[1,2],
Silvia Bianconcini[4], Francesco Maicu[2], and Francesco Trotta[2]
[1] Dept. of Physics and Astronomy, University of Bologna, Bologna, Italy
[2] CMCC Foundation - Euro-Mediterranean Center on Climate Change, Italy
[3] Dept. of Biological, Geological and Environmental Sciences, University of Bologna, Bologna, Italy
[4] Dept. of Statistics, University of Bologna, Bologna, Italy
*Corresponding author*: Mahmud Hasan Ghani, mahmudhasan.ghani2@unibo.it



# Abstract

Understanding the surface heat budget of the Mediterranean Sea is essential for assessing its role in regional climate and ocean circulation. Under the steady-state heat budget closure hypothesis, the Mediterranean should exhibit a net surface heat loss to balance the heat gained through the inflow of warm Atlantic water at the Gibraltar Strait. However, literature estimates of the net heat flux vary widely, raising questions about the accuracy of existing reanalysis products. In this study, we compute the net surface heat flux over the Mediterranean using two atmospheric datasets: high-resolution (0.125°) ECMWF analysis and lower-resolution (0.25°) ERA5 reanalysis. By applying the same sea surface temperature fields and bulk formulas in both cases, we isolate the impact of atmospheric resolution and data quality. We find that the ECMWF analysis yields a basin-averaged net heat flux of $-3.6 \pm 1.3 \, \mathrm{W\,m^{-2}}$, consistent with the closure hypothesis, while ERA5 gives a spurious positive flux of $+5 \pm 1.2 \, \mathrm{W\,m^{-2}}$. Furthermore, beyond simply assessing the net heat budget, this study delves into the probability distributions of air-sea heat fluxes, aiming to gain a deeper understanding of associated uncertainties and extreme values in turbulent heat fluxes. The probability distributions for turbulent heat flux components exhibit characteristics such as skewness and kurtosis, respectively, varying across the basin. To assess the influence of extremes, we apply the Interquartile Range (IQR) method within statistical models that account for the skewed nature of turbulent heat flux distributions, enabling a consistent treatment of outliers. Our results reveal that extreme negative heat flux events play a critical role in determining the net heat flux direction; excluding these extremes leads to a spurious positive heat budget. This highlights the importance of high-resolution atmospheric data for accurately capturing air-sea interactions and ensuring physically consistent climate modelling over the Mediterranean Sea. And we demonstrate that the Mediterranean heat budget closure hypothesis is connected to extreme heat loss events occurring in key regions of the basin, such as the Gulf of Lion, the Adriatic Sea, the Aegean Sea, and the southern Turkish coasts.

KEYWORDS: Heat fluxes, Mediterranean net heat budget, Fluxes probability distributions, heat flux extremes



## 1. Introduction

The exchange of momentum, water, and heat between the atmosphere and ocean plays a pivotal role in connecting their dynamics (Kara et al., 2000). These fluxes, influenced by atmospheric surface variables and Sea Surface Temperature (SST), drive ocean circulation (Large and Yeager, 2009; Small et al., 2019).

Our study focuses on the Mediterranean Sea, a unique semi-enclosed anti-estuarine basin where heat, water, and momentum fluxes intertwine to fuel a robust vertical circulation (Pinardi et al., 2019). We aim to reassess the long term mean net heat flux of the basin since this flux is a source of energy for the basin wide circulation (Cessi et al., 2014). Moreover, the Mediterranean net heat budget comprises of several terms that show a considerable range of uncertainty due to large temporal uncertainties arises from the surface fluxes (Jordà et al., 2017).

Understanding the heat balance in the Mediterranean Sea has long been a formidable task (Bignami et al., 1995; Castellari et al., 1998; Matsoukas et al., 2005; Pettenuzzo et al., 2010; Sanchez-Gomez et al., 2011; Criado-Aldeanueva et al., 2012; Jordà et al., 2017), whether through numerical models or observational data analysis. The fundamental problem of in-situ observations is the limited space-time extension of the data sets, while for numerical modelling, the limitations lie in the semi-empirical approach of the air-sea bulk formulas. Numerous endeavours have been undertaken (Large and Yeager, 2009) to calculate air-sea heat fluxes using atmospheric state variables obtained from in-situ observations, remote sensing data, or numerical model outputs. In our study, we utilize atmospheric analysis and reanalysis data, which provide an optimal reconstruction of past atmospheric surface state variables using models and observations.

Numerous past studies have employed well-established bulk transfer formulas to estimate air-sea fluxes (e.g., Fairall et al., 2003; Pettenuzzo et al., 2010; Cronin et al., 2019). The turbulent heat flux components-latent and sensible heat flux are commonly derived from surface wind speed, sea surface temperature, near-surface air temperature, and humidity (Large and Yeager, 2009). However, Gulev and Belyaev (2012) noted that global heat flux products often vary significantly, mainly due to differences in the bulk formulations and input variables adopted across studies.

At the Gibraltar Strait, the Mediterranean Sea exchanges water with the Atlantic through a characteristic two-layer flow: warm, relatively fresh Atlantic water enters at the surface, while colder, saltier Mediterranean water exits at depth. This arrangement leads to a net gain of heat for the Mediterranean basin, since the incoming surface water carries more thermal energy than the colder outflow. To maintain a long-term heat balance, this lateral heat gain must be compensated by a net loss of heat at the sea surface. In other words, the basin-average surface heat flux should be negative-a constraint known as the heat budget closure hypothesis. Accurately estimating this surface heat flux remains a challenge due to limited data and uncertainties in flux parameterizations. A benchmark estimates of the net heat budget, -7 W m$^{-2}$, was proposed by Béthoux et al. (1998), though it is based on data from the 1970s and 1980s and may not reflect present-day conditions under a changing climate (Criado-Aldeanueva et al., 2012; Marullo et al., 2021).

Recent studies highlight significant uncertainty in the estimated long-term net heat budget of the Mediterranean Sea, with some even reporting positive values. Song and Yu (2017), presented an ensemble climatology of surface heat fluxes, reporting a net heat budget of 2±12 W m$^{-2}$ and noting a warm bias in this ensemble estimate. Utilizing an



ensemble of high-resolution regional climate models (RCMs), Sanchez-Gomez et al. (2011) found that individual
RCMs did not achieve a heat budget closure, but the ensemble mean heat flux was -7± 21 W m⁻². Ruiz et al. (2008)
achieved a heat budget of -1 W m⁻² using downscaled NCEP/NCAR global reanalysis of ½° x ½° resolution, but
their computed heat flux components values have shown large deviation than most of the literature values (the major
difference is in the value of 168 W m⁻² for net short wave downward). Merullo et al. (2021) recently analysed
several atmospheric data sets, revealing a significant net heat flux variability ranging between 1.6 and 40 W m⁻².
They attributed this variability primarily to longwave radiation fluxes uncertainties. In addition to these challenges,
past studies of air-sea fluxes have primarily focused on establishing mean and variance, leaving limited knowledge
about their statistical distributions (Korolev et al., 2015; Tian et al., 2017). Understanding the probability
distributions of air-sea fluxes and their higher moments could provide insights into the uncertainties associated with
air-sea physics. Also, the analysis of probability distributions can help to assess skills of different reanalyses to
replicate extreme fluxes (Gulev and Belayaev, 2012).
In this study we investigate two very different aspects of the net surface heat budget closure problem of the
Mediterranean Sea. First, we employ two different by high quality surface atmospheric variable data sets at different
horizontal resolution and we calculate the heat fluxes with the same bulk formula and the same SST. This isolates
the impact of atmospheric model resolution and quality as the sole source of variation in the heat flux estimates.
Therefore, we answer the question: is the Mediterranean Sea in the past 15 years still losing heat at the surface?
Secondly, we study the statistical distributions of the heat flux components, utilizing the atmospheric analysis dataset
which are used to produce weather forecast by ECMWF (European Centre for Medium-Range Weather Forecasts).
Knowing the skewness and kurtosis distributions across the basin, we analyse the extremes of the net heat budget,
and we determine the specific importance of the extreme heat losses to the long-term mean. The second question
we answer is: what are the underlying causes of the net heat budget closure problem?
The paper is structured into the following sections. Section 2 presents the atmospheric analysis and reanalysis
datasets from ECMWF, along with satellite SST data and the bulk formula used in the estimation of the fluxes. In
Section 3, we present the new values of the heat budget closure problem, compared to the literature. In section 4,
we analyse the probability distributions of turbulent heat fluxes. In Section 5, we determine the causes of the long
term mean net heat budget values. Finally, Section 6 summarizes the findings and highlights key insights gleaned
from this research.

## 2. Methodology and datasets

### 2.1 Air-sea physics in the Mediterranean Sea

For the Mediterranean Sea, several formulations have been established over the past decades through extensive
studies. In this section, we present these adopted formulations, beginning with the net heat flux formula, followed
by the specific heat flux components utilized in this study.



The net surface heat flux, $Q_{net}$ comprises the net shortwave radiation $Q_{SW}$, net longwave radiation $Q_{LW}$, and surface
turbulent flux components, which encompass the latent heat flux of evaporation $Q_{LH}$ and sensible heat flux $Q_{SH}$
(Cronin et al., 2019; Pettenuzzo et al., 2010).

$$Q_{net} = Q_{SW} + Q_{LW} + Q_{lat} + Q_{sen} \tag{1}$$


Here, we use the convention that positive heat fluxes denote heat gain by the ocean. We did not use directly the
atmospheric model heat flux values since we wanted to intercompare two different atmospheric data sets in terms
of their quality and resolution not on the basis of the specific bulk parametrizations and SST used. Thus, we used
the same bulk formula and SST for both ECMWF and ERA5 surface variables that are described in section 2.2.

### 125 2.1.1 Shortwave radiation flux

The shortwave radiation flux (SW) is derived from the formulation proposed by Rosati and Miyakoda (1988). The
largest heat flux component is the solar radiation which is reduced by the cloud coverage and partially reflected by
the sea surface (albedo). The shortwave heat flux formula is therefore expressed as:

$$Q_{SW} = Q_{TOT}\,(1 - 0.62\,C + 0.0019\,\beta)(1 - \alpha) \quad if\ C \geq 3 \tag{2}$$
$$Q_{SW} = Q_{TOT}\,(1 - \alpha) \qquad\qquad\qquad if\ C < 3$$


where $Q_{TOT}$ indicates the clear sky solar radiation calculated with astronomical formulae, C is the fractional cloud
coverage, β is the noon solar altitude in degrees and α is the ocean surface albedo varying month wise values taken
from Payne (1972). The incoming solar radiation varies on locations with sun zenith angel and $Q_{TOT}$ reaches at the
ocean surface after diffusion can be represented by the components: the sum of the direct solar radiation $Q_{DIR}$ for
direct solar radiation and $Q_{DIF}$ for downward diffused radiation. Then net solar radiation $Q_{TOT}$ can be represented
by the summation of components $Q_{DIR}$ and $Q_{DIF}$ :
$$Q_{TOT} = Q_{DIR} + Q_{DIF}$$
$$= Q_0\,\tau^{sec\,z} + [\,(1 - A_a)\,Q_0 - Q_0\,\tau^{sec\,z}] * 0.5$$

Here $Q_0$ is the solar radiation at the top of atmosphere, $\tau$ is equal to 0.7 and is the atmospheric transmission
coefficient, $A_a$ is a constant value (0.09) and z is the sun zenith angle.







### 2.1.2 Longwave radiation flux

The longwave surface radiation flux (LW) is the difference between the upward infrared radiation (IR) emitted from
the ocean surface (LU) and the atmospheric downwelling infrared radiation (LD). The LD component is adapted
from Bignami et al. (1995), and the longwave radiation flux is written as:

$$Q_{LW} = Q_{LU} + Q_{LD} \tag{3}$$
$$Q_{LU} = -\epsilon \, \sigma_{SB} \, T_S^4 \tag{4}$$
$$Q_{LD} = [\, \sigma_{SB} \, T_A^4 \, (\, 0.653 + 0.00535 \, e_A)](1 + 0.1762 \, C^2) \tag{5}$$

where: $T_S$ and $T_A$ indicate the sea surface temperature and air temperature in degrees Kelvin, $\sigma_{SB}$ is the Stefan-
Boltzmann constant, $\epsilon$ is the ocean emissivity set to 1 according to Large and Yager (2009) and $e_A$ is the atmospheric
vapor pressure computed from the mixing ratio of the air $W_{air}$ (Wallace and Hobbs, 2006).
$$W_{air} = \frac{q_A}{1 - q_A} \tag{6}$$
$$e_A = \frac{W_{air}}{(W_{air} + \gamma)} \, p \tag{7}$$
and $q_A$ is the specific humidity of air, $p$ is the surface air pressure, and $\gamma$ is a constant (0.622).
The specific humidity ($q_A$) saturated at the $T_A$ is computed using the following equation (Large, 2006)
$$q_A = \rho^{-1} \, 640{,}380 \exp(\,^{-5107.4/T_D}\,) \tag{8}$$
where, $T_D$ is the dew point temperature retrieved from the atmospheric model outputs.

### 2.1.3 Turbulent heat fluxes

The turbulent heat flux is composed of sensible heat $Q_{SH}$ and latent heat $Q_{LH}$ given by the following formula:
$$Q_{SH} = -\rho_A \, C_P \, C_H \, |\vec{V}| \, (T_S - T_A) \tag{09}$$
$$Q_{LH} = -\rho_A \, L_E \, C_E \, |\vec{V}| \, (q_S - q_A) \tag{10}$$

where $|\vec{V}|$ is the wind speed, $\rho_A$ is the density of moist air, $C_P$ is the specific heat capacity (1005 J g⁻¹·K), $C_H$ and
$C_E$ are turbulent exchange coefficients for temperature and humidity, $L_E$ is the latent heat of vaporization, $q_A$ is
defined in (8) and $q_S$, which is the specific humidity of air saturated at the sea surface temperature $T_S$, is calculated
with (8) using $T_S$ instead of $T_D$, and applying a 0.98 factor (Sverdrup, 1942). Since the average wind speed in the
Mediterranean is 5 m/s, Pettenuzzo et al. (2010) suggested using constant turbulent exchange coefficients such as
$C_H = 1.3 \times 10^{-3}$ and $C_E = 1.5 \times 10^{-3}$.



**2.2 Datasets**

Two atmospheric datasets have been selected for this study. The first dataset is the ECMWF (European Centre for
Medium-Range Weather Forecasts) high-resolution analysis dataset (Rabier et al., 2000) at six-hourly temporal
resolution and 0.125 degrees of spatial resolution. It's worth noting that the original operational dataset, from which
the atmospheric fields have been extracted, underwent changes between 1991 and 2006 in terms of model resolution
and the assimilated number of observations. For consistency, we opted to utilize the dataset with approximately
uniform model resolution and physics spanning a 15-year period from 2006 to 2020. The second dataset employed
in this study is ERA5 reanalysis (Hersbach et al., 2020). This dataset is available at hourly intervals. However, it
features a horizontal resolution of 0.25 degrees.
To mitigate unresolved atmospheric temperature daily cycles in ECMWF and make the two data sets consistent for
the time variability, the ECMWF and ERA5 fields are further processed into daily mean values for the entire period.
Comparisons conducted with daily and six-hourly input fields indicated minimal differences in the probability
distributions of the heat fluxes, leading us to prioritize filtering out daily variability to the greatest extent possible.
To compute the heat fluxes the following surface variables are extracted from the two datasets: the 10-meter wind
components (U for the zonal direction and V for the meridional direction), mean sea level pressure, dew point
temperature, total cloud coverage, and 2-meter air temperature.

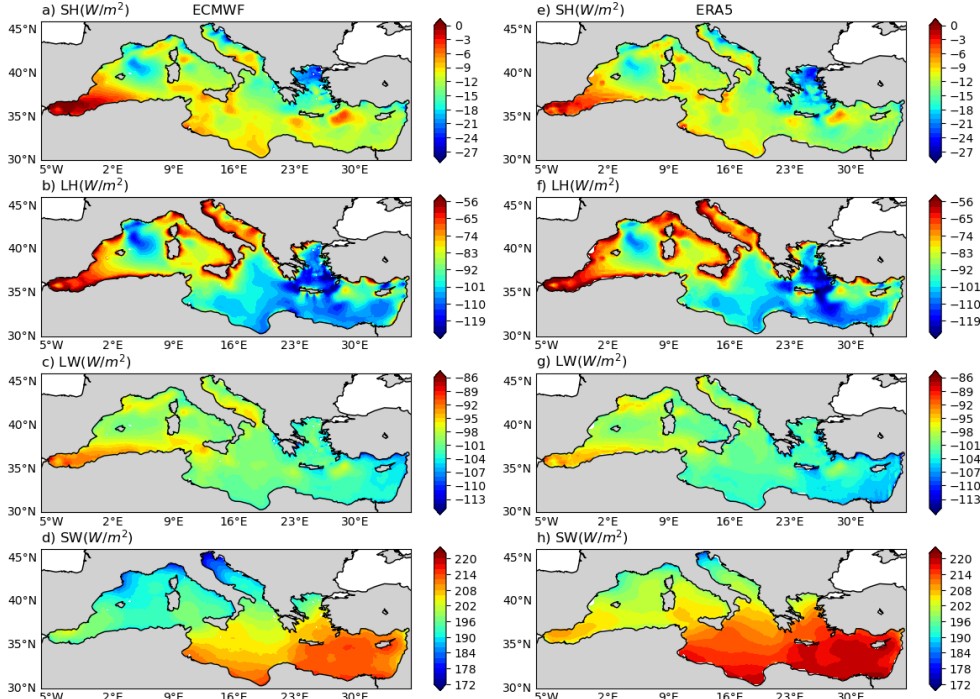

**Figure 1: Mean annual heat flux components for the period of 2006-2020 computed from ECMWF (left) and**
**ERA5 (right) daily time series**



For the oceanic SST data, we utilized the satellite dataset distributed by the Copernicus Marine Environment Service
(CMEMS). This SST dataset is a blended product from multiple satellite sensors, categorized as L4, with a
horizontal resolution of 0.05° × 0.05°. To align the SST data with the atmospheric analysis and reanalysis dataset
grids, we applied an interpolation and extrapolation method known as the 'sea-over land' (De Dominicis et al., 2013).
This method involves an iterative process to extrapolate sea values over land before interpolating, thus not allowing
the contamination of land values on the interpolation.

**3. Heat budget closure problem revisited**

**3.1 Analysis of the heat budget components**


We compute the heat fluxes for the 15-year period, 2006–2020, using the ERA5 dataset and compare them with the
fluxes computed using the ECMWF dataset (Fig. 1). In Fig. 1 we show the results for the 15-year mean of each
heat budget components. We start describing the ECMWF patterns and then we detail the differences.
Turbulent heat fluxes exhibit distinct sub-basin-scale patterns, varying between the eastern and western
Mediterranean Seas as well as the Central Mediterranean region. The largest mean sensible heat loss is observed in
the whole Alboran sea area with absolute value range of 0-6 W m⁻², while the Aegean Sea and the centre of Gulf of
Lion gains more heat in the maximum value of 25 W m⁻². Similarly, the highest absolute values of LH are recorded
in the Gulf of Lion and the Aegean and Levantine Seas, attributed to the influence of strong and cold winds like the
Mistral and Etesian in the north-western and eastern Mediterranean regions, respectively. The eastern
Mediterranean emerges as the region with the highest evaporation, reaching approximately 125 W m⁻² in absolute
value. Notably, along the south-eastern coastline, a wide range of maximum absolute values (102-125 W m⁻²) in
the evaporation is observed. The turbulent heat fluxes show limited differences between the ECMWF and ERA5
datasets.
SW fields show the well-known meridional gradients with larger gradient values arising from the ECMWF dataset.
The mean SW exhibits a gradual decline from the eastern to western Mediterranean, influenced by the variation of
the solar zenith angle with longitudes. The difference in SW radiation between the western and eastern
Mediterranean is suggestive of variations in cloud cover, leading to a larger heat gain in the eastern Mediterranean.
Furthermore, ECMWF and ERA5 different values are connected to different cloud cover. Notably, the northern
Adriatic region stands out with a distinct distribution, suggesting it receives relatively less annual solar radiation
compared to other areas. In contrast, the mean longwave (LW) radiation distribution maintains a relatively
consistent range of absolute values between 86–115 W m⁻² across the entire domain with absolute minimum values
in the Alboran Sea, presumably due to the cold Atlantic water inflow. Overall, while the turbulent heat fluxes show
limited differences between the ECMWF and ERA5 datasets, significant discrepancies are observed in radiative
heat fluxes.



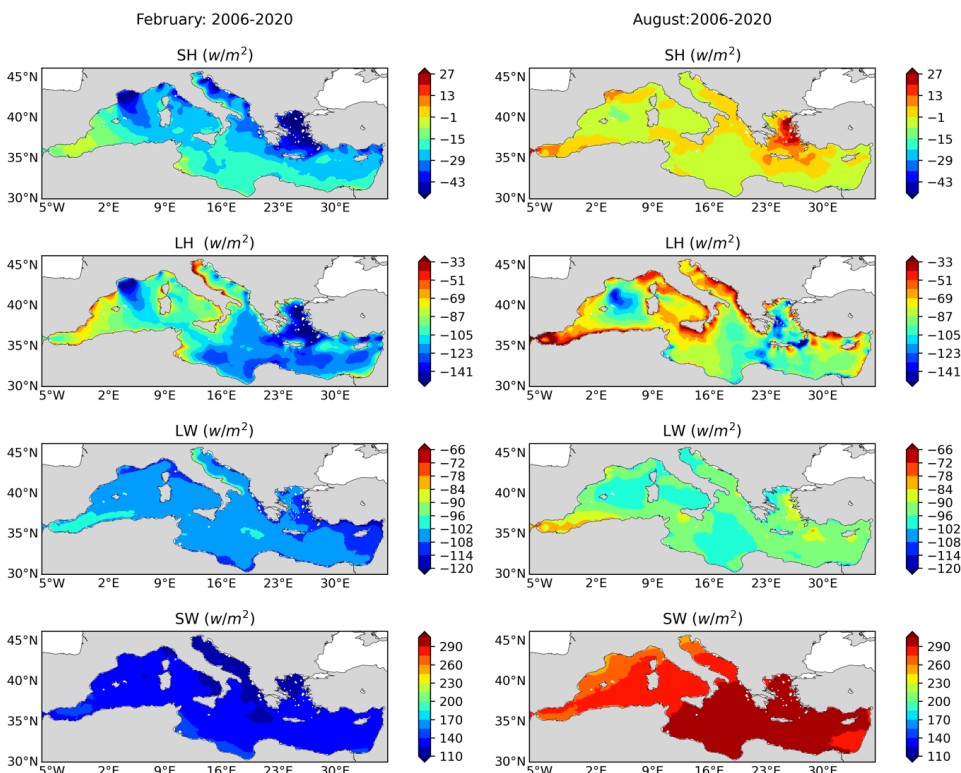

**Figure 2: Seasonal variations of heat flux components: Left column is the monthly average values for February and right column the average for August for the period 2006–2020 (ECMWF data).**

Figure 2 shows seasonal variations in heat flux components for February and August using ECMWF data. Both SH and latent heat (LH) fluxes exhibit a greater spatial gradient in February compared to August. In winter, the SH loss is larger, especially in the Gulf of Lion, Aegean, and parts of the Adriatic, with stronger spatial gradients compared to summer. In August, SH flux becomes positive in the Aegean and the Alboran Sea. LH loss is highest in February in the whole eastern Mediterranean and the Gulf of Lion. In August, LH losses decrease in the western Mediterranean, with absolute value minima in the Alboran and Adriatic Sea, remaining largely negative in the lower part of in the Eastern Mediterranean. SW fields show the strongest seasonal cycle as expected, with the absolute maximum summer of 290-320 W m$^{-2}$ in the Eastern Mediterranean. LW is largest in absolute value in winter showing a small seasonal cycle. Significant seasonal variations are observed in the distribution range for radiative heat fluxes, low in February and high in August across the entire domain. These patterns are quite similar to the ones reported in the literature.





**3.2 Net heat budget estimation**

The net surface heat flux $Q_{net}$ is depicted in Figure 3 for ERA5 and ECMWF and basin-average 15 year mean values are listed together with the literature in Table 1.

Fig. 3 shows that the Gulf of Lion and the Aegean Sea are the areas of maximum heat losses while the basin gains heat in the Alboran Sea, in some areas of the Levantine basin, and in the shelf areas around the Italian peninsula.

The Mediterranean Sea gains comparatively more heat with the ERA5 inputs. Besides the difference in surface domain for $Q_{net}$, for both cases, air-sea flux dynamics is strongly visible in the Alboran Sea for net heat gain, in the Gulf of Lion for heat loss due to the continental cold wind (Mistral wind), and in the Aegean Sea due to the strong wind (Etesian wind) that blows during the summer period. Using ECMWF inputs, $Q_{net}$ is -3.6 W m⁻², a value consistent with previous estimates for the Mediterranean Sea domain and for ERA5, it is 5 W m⁻² (Table 1). Errors in $Q_{net}$ mean value are determined by a bootstrapping method where $Q_{net}$ time series is resampled 5000 times to compute a standard deviation around the mean of the resampled time series (Tibshirani & Efron, 1993). We argue that our results show that the negative heat budget is achieved by using only ECMWF fields at high resolution, i.e. 0.125 degrees. Higher resolution implies differences in all atmospheric fields used to compute the fluxes. Furthermore, ERA5 and ECMWF model physics and dynamics is different contributing to the differences in the mean heat budget. However, both datasets use observations, and we argue that the most relevant difference between the analysis and the reanalysis data set is the resolution due to the peculiar geometry of the Mediterranean Sea.

Since all the literature datasets are coarser, this is most likely the reason of the failure to determine the correct heat budget closure value. In Pettenuzzo et al. (2010) several ad-hoc corrections were made to the surface atmospheric fields to obtain the negative heat flux budget while in Sanchez-Gomes et al., (2011) they used an ensemble of deterministically downscaled ERA40 fluxes giving rise to a very large uncertainty. Considering a recent literature, our resulted $Q_{net}$ is closely matched with the computed net heat budget of -3±8 W m⁻² from Jordà et al., (2017), but their result was associated to large temporal uncertainties from the surface fluxes through Gibraltar Strait.

Thus, comparing the net heat flux $Q_{net}$ estimates derived from ERA5 reanalysis and ECMWF analysis dataset, using specific bulk formula demonstrate an uncertainty in the results. This uncertainty, potentially linked to spatial resolution difference between the two datasets, impacts the regional heat budget closure. We are now able to answer the question: is the Mediterranean Sea in the past 15 years still losing heat at the surface? The answer is yes, notwithstanding the climate change warming, the Mediterranean still looses heat due to its heat budget closure constraint. However, it might be that it looses less than in the previous decades, and this will be the scope of more studies in the future.




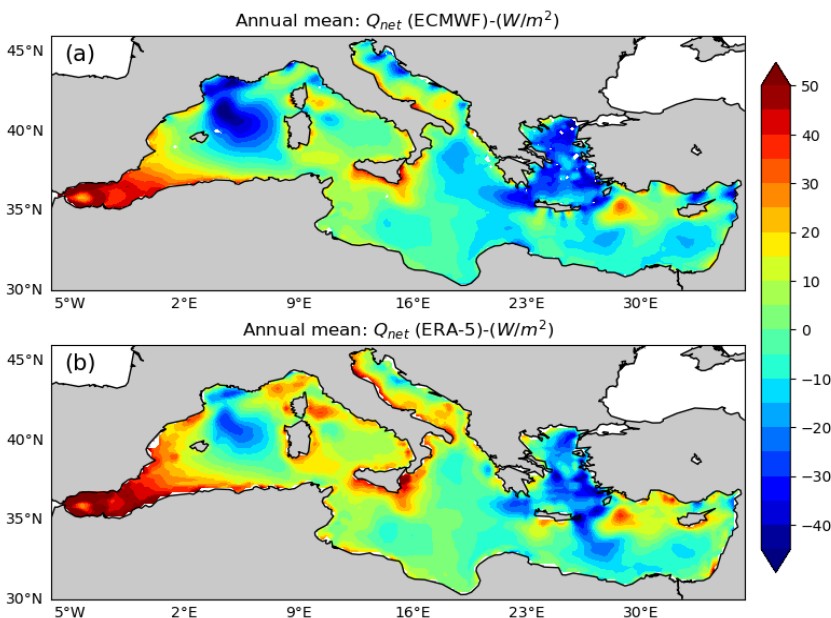


**Figure 3: Comparison of the annual $Q_{net}$(W m⁻²) computed from, a) ECMWF and b) ERA-5 input datasets.**

**Table 1: Computed flux components and net heat fluxes ($Q_{net}$), and values from the references**

| Authors | SH | LH | LW | SW | Net Flux ($Q_{net}$) |
|---|---|---|---|---|---|
| Bethoux (1979) | -13 | -120 | -68 | 195 | -6 |
| Bunker (1982-1) | -13 | -101 | -68 | 202 | 20 |
| Bunker et al (1982-2) | -11 | -130 | -68 | 202 | 20 |
| May (1986) | -11 | -130 | -68 | 193 | 2 |
| Garret et al. (1993) | -7 | -99 | -67 | 202 | 29 |
| Matsoukas et al. (2005) | -11 | -122 | 186 | -63 | 22 |
| Ruiz et al. (2008) | -8 | -88 | -73 | 168 | -1 |
| Pettenuzzo et al. (2010) | -14 | -90 | -79 | 178 | -7 |
| Sanchez-Gomez et al. (2011) | -13±5 | -100±13 | -75±6 | 181±18 | -7±21 |
| Criado-Aldeanueva et al. (2012) | -15.1 | -93.5 | -76.9 | 186.3 | 0.73 |
| Song & Yoy (2017) | -13±4 | -98±10 | -78±13 | 192±19 | 2±12 |
| Jordá, et al., 2017 | - | - | - | - | -3±8 |
| ECMWF analyses | -12.1±4 | -92±16 | -100.5±3 | 201±8 | -3.6±1.3 |
| ERA5 reanalysis | -13±3 | -89±14 | -101±3 | 208±8 | 5±1.2 |



Spatially, the mean $Q_{net}$ distribution generally shows a heat loss across much of the Eastern Mediterranean. Overall,
distributions of more positive net heat budget values for the western Mediterranean and negative for the eastern
Mediterranean have matched with the similar result from Criado-Aldeanueva et al. (2012). Strong spatial gradients
are evident, particularly in the Aegean Sea, although a few patches displaying net heat loss (negative $Q_{net}$) are also
noticeable in this vicinity. Conversely, the western Mediterranean exhibits a stronger heat gain area, which appears
particularly concentrated zone in the Gulf of Lion region and this feature is apparent in results from both atmospheric
datasets. Such a spatial related uncertainty in $Q_{net}$ represents a significant challenge for accurately closing regional
heat budgets as well as validating existing ocean circulation models within the complex Mediterranean basin.

**4. Probability distributions of the turbulent heat fluxes**
In this section, we analyse the probability distribution of turbulent heat fluxes computed using ECMWF data set
only and for the anomaly heat fluxes.
If we indicate the time series of each component of the heat budget with $X_n$ we can define the heat flux climatology
as:

$$Q_t = \frac{1}{n} \sum_{j=1}^{n} X_{tj} \tag{11}$$

where 't' indicates the day of the year and ' j' is the number of years. The anomaly time series is computed by
subtracting the long-term seasonal climatology $Q_t$ from the observed heat flux time series $X_{tj}$ and it will be indicated
by:

$$\tilde{X}_{tj} = X_{tj} - Q_t \tag{12}$$

**4.1 SH flux distribution**
We found that gaussian or skew-normal distributions are not well fit for SH flux, as evident from the histograms at
single grid points shown in Figure 4a. The histograms reveal a singularity around zero, indicating that the skew-
normal distribution may not adequately capture the distribution of these values. This observation is consistent with
findings by Gulev and Belyaev (2012), and we provide further explanation in the Appendix A.
The most common distribution with such near-discontinuous behaviour at the origin is the three-parameter
Asymmetric Laplace Distribution (ALD) (Yu & Zhang, 2005) that we can defined as
$$F(x, \alpha, \mu, \lambda) = \frac{\lambda}{\alpha + \frac{1}{\alpha}} \begin{cases} \exp\left(\frac{\lambda}{\alpha}(x - \mu)\right) & \text{if } x < \mu \\ \exp\left(-\lambda\,\alpha(x - \mu)\right) & \text{if } x \geq \mu \end{cases} \tag{13}$$
where x is the random variable time series, α is the shape parameter, μ is the location and λ the scale.





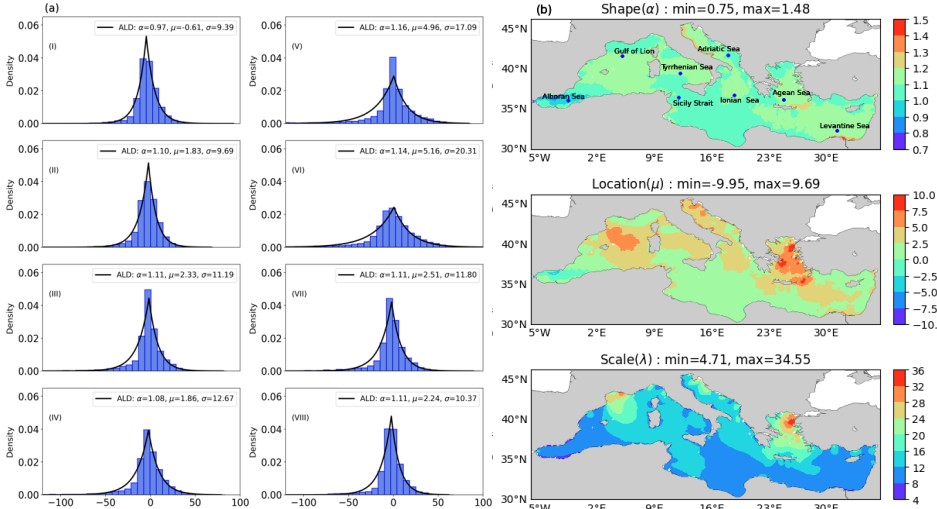

**Figure: 4 a) The single grid point histograms for SH flux anomalies from the eight sampling locations for the period of 2006-2020, 4 b) The Asymmetric Laplace PDF parameter (α, μ, λ) distributions from computed SH flux anomaly for the observation period. [Sampling points: (I) Alboran Sea, (II) Gulf of Lion, (III) Tyrrhenian Sea, (IV) Sicily Strait, (V) Adriatic Sea, (VI) Ionian Sea, (VII) Aegean Sea, (VIII) Levantine Sea]**

From the single grid point histogram, we have observed a one or two sharp peaks in the distribution that matches well with the Asymmetric Laplace Distribution (ALD) PDF. In accordance with findings by Yu and Zhang (2005), the distribution of the sensible heat (SH) flux anomaly time series exhibits characteristics of a double exponential distribution. This is evident from the histograms displaying both positive and negative skewness with long tails, as depicted in Figure 4a. The ALD parameters for the SH flux anomaly time series are illustrated in Figure 4b. The shape parameter (α) falls within the positive range of 0.73 to 1.48, indicating a moderate to high degree of peakiness in the distribution. Additionally, the location parameter (μ) exhibits both positive and negative values, suggesting a shift in the central tendency of the distribution. Notably, the scale parameter (λ) displays a similar structure to the SH flux climatology depicted in Figure 1.

To check the quality of the fit, moments of both applied and theoretical PDF are compared (presented in supplementary materials, Fig. S3). The comparison shows the estimations of the three moments in the left panel for the observed SH flux and right panel for ALD PDF parameters. It can be seen that variances and skewness are similar in distribution while kurtosis differ at noticeable range. This observation is likely attributed to the fact that the kurtosis for the asymmetric Laplace distribution remains constant regardless of changes in the scale parameter.

**4.2 LH flux distribution**

In the case of the LH flux, no sharp exponential peaks were observed; instead, large skewness and long tails were identified. Therefore, we applied the skew-normal PDF which is defined by α (∈ R) as the shape parameter, μ (∈ R) the location parameter, and λ >0 the scale parameter (Azzalini, 1985) and defined as:



$$f(x, \alpha, \mu, \lambda) = \frac{2}{\lambda} \phi(\frac{x-\mu}{\lambda}) \Phi(\alpha \frac{x-\mu}{\lambda})$$ (14)


Where,
$$\phi\left(\frac{x-\mu}{\lambda}\right) = \frac{1}{\sqrt{2\pi}} e^{-\frac{1}{2}\frac{(x-\mu)^2}{\lambda^2}}$$ (15)

$$\Phi\left(\alpha \frac{x-\mu}{\lambda}\right) = \int_{-\infty}^{\alpha\frac{x-\mu}{\lambda}} \phi(t)dt$$ (16)

A skew-normal PDF is an extension of the normal distribution while covering the skewness and containing the
general characteristics of a Gaussian distribution (Flecher et al., 2010).
To examine visually the quality of PDF fit on LH flux anomaly values, histograms from eight sea locations were
fitted with the skew-normal PDF, as shown in Figure 5a. Figure 5b displays the parameter spatial variability. The
shape parameter distribution ranges from -6.83 to -2.5, with negative values observed across all points. This spatial
distribution of α, exhibiting a negative range, aligns with the negatively skewed pattern identified in the single grid
point PDF fitting test. Furthermore, the spatial distribution structure of the location and scale parameters
demonstrates a positive correlation across most locations.
In the Supplementary material, a comparison of statistical moments is conducted to qualitatively validate the fit
(supplementary materials, Fig S4). There is notable agreement in the variance distributions between the observed
LH flux anomaly and skew-normal PDF. While the skewness distributions mismatch at negligible level, with the
theoretical PDF skewness predominantly ranging from -0.9 to -0.3, whereas the observed skewness exhibits a
variation range spanning from -1.2 to over -0.3. Lastly, the kurtosis distribution of the skew-normal PDF differs in
the Aegean Sea, Alboran Sea and Gulf of Lion area.

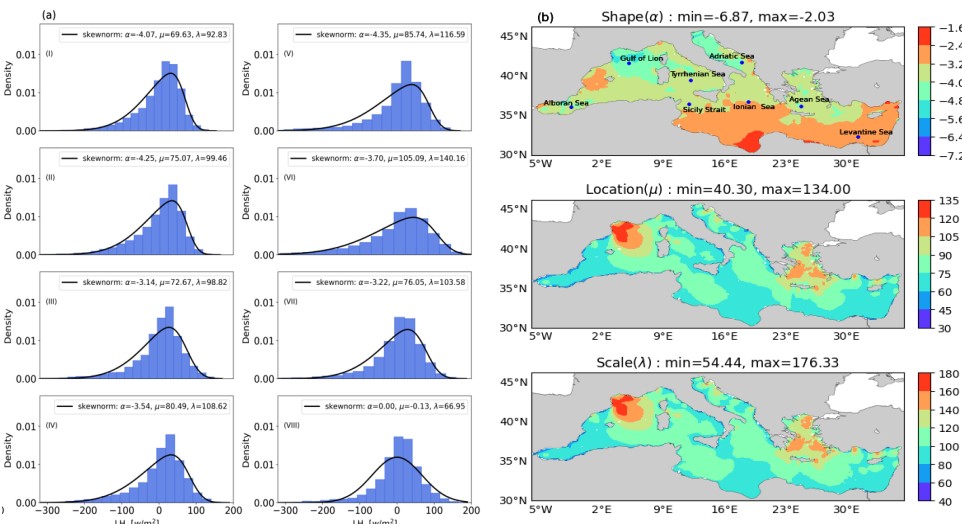

**Figure 5 a) The single grid point histograms for LH flux anomalies at the eight sampling locations for the**
**period of 2006-2020, 5 b) The skew-normal PDF parameter (α, μ, λ) distributions for computed LH Flux**
**anomaly for the observation period [Sampling points: (I) Alboran Sea, (II) Gulf of Lion, (III) Tyrrhenian**
**Sea, (IV) Sicily Strait, (V) Adriatic Sea, (VI) Ionian Sea, (VII) Aegean Sea, (VIII) Levantine Sea]**



### 4.3 Evaluation of the PDF fitting

In this section, we conducted a goodness of fit test to measure the distance between the empirical observed distribution and the fitted one. The objective of this evaluation test was to assess the degree of agreement between the applied theoretical distribution and the observed time series. The chi-squared method, a well-accepted test, was employed to measure the distance between two independent distributions.

We compared the results of the chi-squared test for the turbulent heat fluxes computed using the ECMWF and ERA5 datasets. The decision rule for the χ2 test was determined based on the level of significance, set at 0.05, and the degrees of freedom, defined as DF = N - np, where N represents the number of bins and np is the number of distribution parameters (i.e., 3 for both the ALD and skew-normal distribution). In the supplementary material we show the maps of Chi-square test statistics (Fig. S5). The chi-squared results for the SH and LH fluxes computed using the ECMWF dataset indicate that almost all surface grid points are well-fitted with the applied theoretical PDFs, the ALD and skew-normal PDF. With the critical threshold of 33.92 (Elderton, 1902) for P values, we observed a very few mismatches location near the coasts.

### 5. How do heat loss extremes contribute to the heat budget closure?

The heat budget closure problem is associated with achieving a net negative heat flux, as discussed before. We test here the hypothesis that the negative long term mean negative heat budget of Table 1 for ECMWF data is correlated to the extremes in heat losses during autumn-winter.

Figure 6 illustrates the $Q_{net}$ basin average daily time series, revealing a value range varying between 200 and -500 W m$^{-2}$. Notably, the most pronounced extreme negative heat loss peaks, reaching up to 500 W m$^{-2}$ occur in the winters of 2011, 2015 and 2017. They approximately coincide with western Mediterranean Deep Water formation events, as documented in Escoudier et al. (2021). To identify and remove the potential extremes in our computed $Q_{net}$ time series, we apply the Interquartile Range (IQR) method which measures the spread of a dataset and calculate the difference between the third quartile(Q3) and the first quartile (Q1). The IQR threshold is computed by the difference between the 1$^{st}$ quartile (Q1) and 3$^{rd}$ Quartile (Q3) of the observed dataset:

$$IQR = \ Q3 - Q1 \tag{17}$$
$$Threshold = Q1 - k * IQR \tag{18}$$

We used different values for k to exclude the negative extremes, which correspond to the maximum heat losses. These extremes values are replaced with long term yearly climatology values for the extreme heat losses days and the long term mean neat heat budget $Q_{net}$ is recomputed.

The resulting $Q_{net}$ for different thresholds is displayed in Table 2 and the thresholds are shown in Fig. 7 together with the daily seasonal climatology. If compared with the long term mean heat budget in Table 1 (-3.6 W m$^{-2}$) we see that eliminating the winter extremes produces a smaller long term mean heat loss up to changing the sign to



positive values. We argue that this is reason why ECMWF and ERA5 have so different $Q_{net}$ long term mean values, the low resolution of ERA5 does not allow for extreme heat losses in the winter. Furthermore, if we calculate the yearly mean value of the seasonal climatology, we obtain the value of +4 W m$^{-2}$, which confirms again the importance of extremes in the heat budget closure of the Mediterranean Sea.

The $Q_{net}$ could become an impact indicator of the Mediterranean steady state balance and be used to see if trends are working on the overall basin.

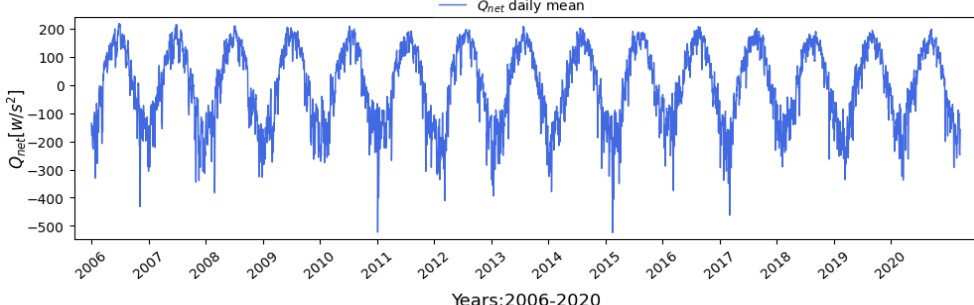

**Figure 6: Basin averaged time series of the computed daily $Q_{net}$ (units W m$^{-2}$) from the ECMWF computed heat fluxes, for the period 2006-2020.**

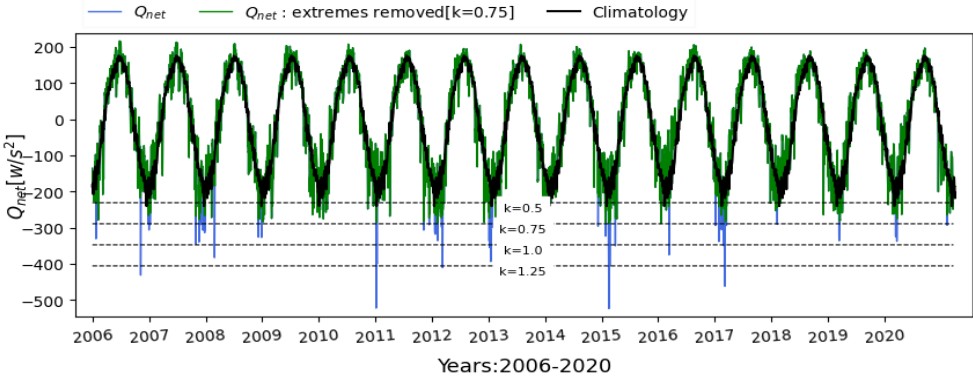

**Figure 7: Time series of the basin averaged $Q_{net}$ means, $Q_{net}$ extremes removed, and long-term yearly climatology and four lower quantile boundary line marked with dashed lines using different k values [k=1.25, 1,0.75, 0.5].**




**Table 2: Different lower quantile boundary limits used to replace potential extremes and the resulting long-term mean basin averaged $Q_{net}$ values**

| IQR lower boundary limit | Threshold value (W m⁻²) | $Q_{net}$(Wm⁻²) |
|---|---|---|
| K=1.25 | 405 | -3.2 |
| K=1.0 | 347 | -2.5 |
| K=0.75 | 289 | -1 |
| K= 0.5 | 231 | 2 |


The new spatial long-term means heat budget structure with the extremes removed choosing a threshold
of 289 (for k=0.75) is presented in the figure 8, which indicates extremes influence most the net heat
budget structures in the Mediterranean Sea (in comparison with Fig. 3a).  Furthermore, the elimination
of extremes indicates that the net multi-year average of the heat flux in the Mediterranean Sea is connected
to extreme events occurring in the Gulf of Lion, the eastern Adriatic Sea and the southern Turkish shelves.

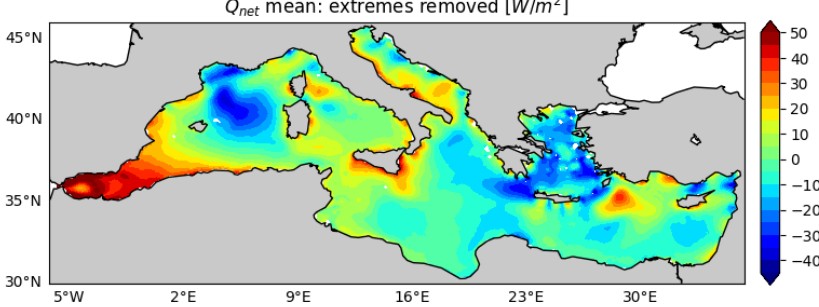

**Figure 8: The annual mean after the removal of extremes with significant reduction of negative heat fluxes**
**in the Gulf of Lion, Adriatic Sea and Aegean Sea regions.**










**6. Discussion and conclusions**

The primary motivation behind this investigation is to revisit the heat budget closure hypothesis from atmospheric consolidated data sets that are nowadays used frequently to drive ocean models. For this analysis, we covered a 15-year period from 2006 to 2020 with a daily time series frequency. The reason for the choice of this time range is that ECMWF analyses became quite stable starting from 2006 while before the model was at coarse resolution, like ERA5's model. Our strategy is to use the same bulk formula for the ERA5 and ECMWF data sets and compute the long term mean heat budget. Our strategy is to use the same SST and the same bulk formula but different atmospheric reanalysis and analysis surface variable data sets and compare the value of the long term mean heat budget in the Mediterranean Sea.

Firstly, the heat budget of the Mediterranean Sea was analysed to examine average annual mean and seasonal variations. The largest component of the heat budget is the net solar radiation (SW), followed by the latent heat (LH), longwave radiation (LW), and then sensible heat (SH), as shown in the literature. All heat flux components exhibit significant seasonality, as illustrated in Figure 3. Differences appear in the structure of the fluxes when different atmospheric data sets are used, a conclusion aligning with a suggestion from Marullo et al. (2021) on the sensitivity of LW estimates from the atmospheric dataset used to calculate fluxes.

The basin-average net heat flux, $Q_{net}$, was calculated to be -3.6±1.3 W m⁻² for ECMWF analysis data while it is 5±1.2 W m⁻² for ERA5 (Table 1). This finding supports the conclusion that heat budget closure hypothesis cannot be satisfied with a relatively coarse reanalysis atmospheric data set. Our initial question was: is the Mediterranean Sea in the past 15 years still losing heat at the surface? The answer is yes if we use a high-resolution ECMWF atmospheric analysis.

Furthermore, we have demonstrated that the probability density of surface heat fluxes can be modelled and fitted with a three-parameter PDF composed of a shape, a location, and a scale parameter. All the turbulent heat flux components show asymmetric behaviour. There is encouraging agreement between the first two statistical moments of the fitted PDF and the observed values. Kurtosis does not seem to be properly captured by the PDF used but our time series is too short to arrive at a definitive conclusion. For the SH we demonstrate that the ALD PDF is generated by the contributing distributions of wind speed (Weibull) and temperature difference, combined to form the heat flux. We believe this is the first time that such kind of transformation is demonstrated.

Gulev and Belyaev (2012) applied the two-parameter Fisher–Tippett distribution (also known as the Gumbel distribution) to monthly sensible and latent heat fluxes derived from NCEP–NCAR reanalysis fields. Their approach focused on using the mean and standard deviation to estimate the distribution's location and scale parameters relevant to extreme events. However, the Gumbel distribution has a fixed skewness, limiting its ability to capture the contribution of rare, asymmetric extremes. In contrast, our study analyses anomalies from the seasonal cycle using full probability distributions that allow for variable skewness. This better reflects the nature of atmospheric and oceanic variables, which are often inherently skewed (Sardeshmukh and Penland, 2015), and is essential for understanding the influence of extremes on the surface heat budget. Our findings show that incorporating a shape parameter is key to accurately capturing distribution structure and preserving asymmetric



tails. This analysis provides a useful framework for validating surface flux products and assessing their variability,
particularly important given that surface fluxes are the dominant source of uncertainty in the Mediterranean net heat
balance (Jordà et al., 2017). Correctly estimating skewness is crucial, as a small number of extreme outliers,
especially during intense winter events, can disproportionately affect the basin-wide mean and determine whether
heat budget closure is achieved.
For the first time, we have investigated the effects of extreme heat losses in the Mediterranean Sea in the long term
mean basin averaged heat budget. The northern basin areas are the site of the largest heat losses (Gulf of Lion and
the Aegean Sea, Adriatic Sea and the Turkish southern coasts). Exclusion of the negative extremes in these areas
resulted in a change in the sign of long term mean heat loss. The threshold value that produces a positive basin
mean heat loss is 231 W m$^{-2}$. Thus, if the basin mean heat loss does not exceed this value, the basin is not in steady
state. This might be a good indicator of Mediterranean Sea heat content trends to be exploited in the future. Our
second initial question was: what is the cause of the Mediterranean Sea negative long-term mean heat budget? The
answer is that the long-term mean, basin averaged heat loss is due to winter extremes in the Northern regions of the
Mediterranean Sea.
In conclusion, understanding the characteristics and distributions of air-sea heat fluxes are crucial for gaining
insights into variations in the heat budget. Moreover, PDFs of heat fluxes that have been estimated in this study
will allow in the future to understand the importance of extreme events to compose the net negative heat budget.
The next steps could involve a machine learning study of air-sea flux bulk parametrizations for different atmospheric
data sets and coupled models, using as target the data set from this study.
















**Author contributions**

MHG: development of the concept, literature review, writing, methodology, coding, formal analysis, wiring, visualization. NP: conceptualization, review, writing, methodology. AN: conceptualization, writing, review. LM: review, writing. SB: methodology, review. FM: methodology, review. FT: methodology, coding.

**Acknowledgments.**

This research has been funded from the University of Bologna under the Future Earth, Climate Change and Societal Challenges PhD Program for MHG. Additionally, there was a partial support from the Edito-Lab HEU project for MHG and NP.

**Conflict of interest Statement**

The authors declare no conflicting interests.



**APPENDIX A**



Here, we show that the characteristics of the SH flux distribution are due to the specific form of the heat flux as
given by (10), i.e. a multiplication of two distributions, wind speed and temperature differences, that are described
by a different distribution.

Let's indicate with P(v*DT) the combined SH distribution of Q(v) for wind speed and R(DT) the temperature
difference as in equation (09). Assuming that the two distributions are independent, the combined distribution is
the product of Q and R. If we now define the variable z=v*DT, the new combined distribution on the heat flux
variable z is given by the Mellin transform and convolution, described in Papoulis, A., & Pillai, S. U. (2002).

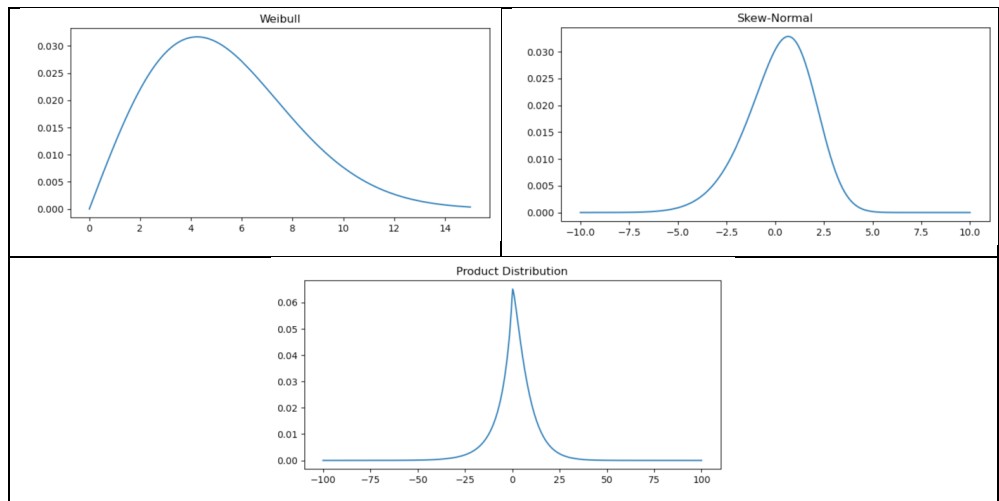

Figure A1: Histograms presenting the two original distributions, Q(v) (upper left quadrant, units wind speed) and
R(DT) (upper right quadrant, units degrees C) and the combined distribution for SH flux in units of $W\,m^{-2}$. The
parameters used for the two original distributions are: k = 2.0 for the Weibull shape, lambda = 6.0 for the Weibull
scale; alpha = - 2.0 for the Skew Normal shape, mu = 2.0 for the Skew Normal location and omega = 2.5 for the
Skew Normal scale






541                                              **APPENDIX B**


The statistical moments for the skew-normal PDF are given by:

$$E(x) = \mu + \lambda\,\delta\,\sqrt{\frac{2}{\pi}} \tag{B1}$$


$$\sigma^2 = \lambda^2\left(1 - \frac{2\delta^2}{\pi}\right) \tag{B2}$$


$$\mu_3 = (4 - \pi)\,\frac{(\delta.\sqrt{2/\pi})^3}{2\,(1-2\delta^2/\pi)^{3/2}} \tag{B3}$$


$$\mu_4 = 2\,(\pi - 3)\,\frac{\left(\delta\,\sqrt{\frac{2}{\pi}}\right)^4}{\left(1-\frac{2\delta^2}{\pi}\right)^2} \tag{B4}$$

where $\delta = \frac{\alpha}{\sqrt{1+\alpha^2}}$. Since the expected value of the time series is zero, we deduce that:

$$\mu = -\lambda\,\delta\,\sqrt{\frac{2}{\pi}} \tag{B5}$$

In other words, location and shape parameters have opposite signs since the scale parameter, $\lambda$, is always positive.
SH flux anomaly distribution was analysed with ALD PDF and its' statistical moments are given by:

$$\text{mean} = \mu + \frac{1-\alpha^2}{\lambda\alpha} \tag{B6}$$


$$\text{variance} = \frac{1+a^2}{\lambda^2\alpha^2} \tag{B7}$$


$$\text{Skewness} = \frac{2\,(1-\alpha^6)}{(\alpha^4+1)^{\frac{3}{2}}} \tag{B8}$$


$$\text{Kurtosis} = \frac{6\,(1+\alpha^3)}{(1+\alpha^4)^2} \tag{B9}$$












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
