# Peer review of "Revisited heat budget and probability distributions of turbulent heat fluxes in the Mediterranean Sea"

_EGUsphere, 2025_

## Author Comment (AC2)

Reviewer comments are in black, answers from the authors are in blue and corrections added to the manuscript are in green

**Anonymous reviewer 1**

Summary: The manuscript investigates several aspects of heat flux dynamics in the Mediterranean Sea. First the manuscript compares the long-term mean heat flux from two reanalysis products with different spatial resolutions (ERA5 and ECMWF) and the higher resolution reanalysis (ECMWF) is found to provide a heat flux consistent with the 'closure hypothesis'. The authors attributed this difference to spatial resolution. Then the authors look at the PDF of the turbulent heat fluxes and also look at the impact of extreme events on the heat budget. The authors find reasonable PDFs that capture the statistical patterns in the heat fluxes terms and that fall/winter cooling events are criteria to achieving the negative long-term mean of ECMWF that is consistent with the 'closure' hypothesis.

The work is interesting and will likely be of interest to a broad range of scientists. Couple of questions that I think should be address in some fashion.

Authors: we thank the reviewer for the interest in our work, and we will try to answer all the questions posed.

**Major comments**

Seems like there's an issue with using the closure hypothesis as evidence for which result is 'correct'. Finding data/results that fit the expectation assumes the hypothesis is truth which seems a bit problematic to me. It seems like the authors should be framing the results differently

We base the closure hypothesis on the heat and mass conservation equations for the Mediterranean Sea. These equations are fundamental for all the earth system modelling; they are not a specific dynamical balance. We argue that, since Gibraltar unequivocally brings heat and water in, the surface fluxes should in principle balance this input. How much is the balance it is not known, and we have hypothesized it is perfect; this is clearly an approximation. As in the seminal work of Bryden and Kinder (1991) and the recent work by Cessi et al. (2014) and Jorda' et al. (2017), the balance between volume integrated heat and mass content helps to understand the basin dynamics. Specifically, being heat entering laterally the Mediterranean Sea, then we search for a negative net surface heat flux. How negative we do not know but searching for a negative net heat flux is a conservative assumption aligned with current scientific understanding.

We have now modified the introduction at line 60:

Furthermore, the estimate of the Mediterranean Sea heat budget from ECMWF meteorological analysis data sets has not been done before.

After line 76 we discuss the hypothesis behind the "closure of the heat budget":

We realise that assuming perfect balance between lateral and vertical heat fluxes, even in the Mediterranean Sea, is an approximation. Being heat clearly entering the Mediterranean Sea through Gibraltar, we search for a negative net heat flux, which we call the closure hypothesis.

How negative such net heat flux is, we do not know but searching for a negative value is a conservative assumption aligned with current scientific understanding.

**References**

- Bryden, H. L., & Kinder, T. H. (1991). Steady two-layer exchange through the Strait of Gibraltar. Deep Sea Research Part A. Oceanographic Research Papers, 38, S445-S463.
- Cessi, P., Pinardi, N. and Lyubartsev, V., 2014. Energetics of semi enclosed basins with two-layer flows at the
  591 strait. Journal of physical oceanography, 44(3), pp.967-979, https://doi.org/10.1175/JPO-D-13-0129.1
- Jorda', G., Von Schuckmann, K., Josey, S. A., Caniaux, G., Garc.a-Lafuente, J., Sammartino, S., ... & Mac.as, D. 624 (2017). The Mediterranean Sea heat and mass budgets: Estimates, uncertainties and perspectives Progress in Oceanography, 156, 174-208 https://doi.org/10.1016/j.pocean.2017.07.001
* * *
It's not clear that the spatial difference in the reanalysis products is the only potential cause of the difference. Could there be other possible causes? For example, could the cloud physics be the issue. Looking at Figure 1, the long wave and short-wave radiation are the terms that look most different. Since SST is the same in both cases, the differences in long wave radiation could point to the cloud cover parameters, maybe? Is cloud representation done the same way in these two reanalysis products. It is not clear to me the only difference in this reanalysis products is the spatial resolution. Maybe I missed that though.

We agree that spatial resolution may not be the only factor causing the difference. Our method allows to eliminate the SST as a possible cause, as noted by the reviewer. We agree with the reviewer that the distribution of cloud cover is also an important difference (reported in Fig A1 below). In fact, the difference is a complex function of different quality of the atmospheric variables. To be noted is that ERA5 and ECMWF analyses use approximately the same model and data assimilation systems. However, we have added also the consideration of cloud cover among the potential differences.

We would like to point out that we had already a sentence at line 223: "Furthermore, ECMWF and ERA5 different values are connected to different cloud cover."

Figure A1: Total cloud coverage (%) mean computed for the period 2006-2020, a) ECMWF and b) ERA-5 input datasets.

To make a more balanced statement we have eliminated in the abstract the sentence at line 35 "This highlights the importance of high-resolution atmospheric data for accurately capturing air-sea interactions and ensuring physically consistent climate modelling over the Mediterranean Sea."

replacing it with:

"Only ECMWF fields are consistent with the heat budget closure hypothesis."

\_\_\_\_\_\_

Why are the statistical distributions just the turbulent heat fluxes explored. Its fine, but it seems like the author should comment on why these are the target and why the distributions of the longwave and shortwave radiation are not explored.

We thank the reviewer for pointing out this missing justification. We wanted to compare our turbulent flux distributions with similar work in the literature which uses only turbulent heat flux components like Gulev and Belyaev (2012) and Korolev et al (2015), also listed below.

- Gulev, S. K., & Belyaev, K. (2012). Probability distribution characteristics for surface air—sea turbulent heat fluxes over the global ocean. Journal of Climate, 25(1), 184-206, https://doi.org/10.1175/2011JCLI4211.1
- Korolev, V., Gorshenin, A., Gulev, S., & Belyaev, K. (2015). Statistical modeling of air-sea turbulent heat fluxes by finite mixtures of Gaussian distributions. In International Conference on Information Technologies and Mathematical Modelling (pp. 152-162). Springer, Cham, https://doi.org/10.1007/978-3-319-25861-4\_13

Furthermore, if we look at the time series of the fluxes (Fig. A2 below), it is clear that the turbulent components are the one that exhibit larger anomalies with respect to the seasonal cycle, hinting to the presence of skewness and kurtosis in their distribution.

Figure A2: Time series of the basin averaged SH, LH, LWV an SWV and their respective extremes removed, overlapped with long-term yearly climatology. Lower quantile boundary lines are marked with dashed lines with different k values used to remove and replace with climatology values

We have now changed the beginning of the section 4 adding, after line 295, the following statement and Fig. A2 in the supplementary material:

Recent studies by Gulev and Belyaev (2012) and Korolev et al. (2015) have analysed the statistical distributions of turbulent heat fluxes, and their findings are used here for comparison. Radiative flux components are excluded from this analysis, as they do not exhibit extremes of comparable magnitude to those of turbulent fluxes (Supplementary Material, Fig. ??). This suggests low skewness and kurtosis in their distributions, reducing the relevance of a detailed probability density function analysis for these components.

\_\_\_\_\_

With regard to the extreme events. I guess it should not be surprising that if you remove the most extreme negative values and then average the heat flux the mean will get warmer. However, potential feedback may not be accounted for that should be mentioned. For example, if the extreme heat flux events are removed but there impacts on SST is not, then the subsequent fluxes may be lower than should be if the extreme event in the heat flux never occurred. So, the relative gain associated with the extreme event may not be as significant indicated by just removing the extreme heat flux. There may be other potential feedback that the removal methodology does not full consider that should be mentioned.

Thanks for raising your concern here. We wanted to show the impact of extreme values on the net heat flux time series because it has never been done before, and it highlights where the major uncertainties in the heat budget closure reside. We understand that in a fully coupled atmosphere-ocean system the feedback can be important but our is a diagnostic study and we cannot change arbitrarily the SST. We do not think we need to comment this in the text but if requested, we can add a comment saying that this is not like to do a simulation changing the atmospheric forcing fields extremes which in turn will generate a different SST.

\_\_\_\_\_

Can more details or discussion be provided on why spatial resolution limits the ability of ERA5 to represent extreme heat loss in fall and winter?

Thanks for the question. A higher spatial resolution is important for capturing many small-scale atmospheric and oceanographic features. In contrast, ERA5 comes with a horizontal resolution of ~31 km, which can smooth many small-scale features and underestimate frequency of extreme fluxes, such extremes are often associated with local wind flows, sharp air-sea temperature contrast and coastal orographic effects.

We apologize for the mistake; we displayed a picture of the fluxes (Fig. 1 and 2) with plot smoothing which did not help to see the noisiness of ERA5 with respect to ECMWF. Here we show an updated Figure 1 and 2 where the differences in resolution are evident in addition to other issues.

We have now replaced Fig. 1 and 2 with the new ones, unsmoothed pictures, and commented in the text after line 229:

Fig. 1 shows the noisiness of the fluxes due to the ERA5 low resolution with respect to ECMWF while retaining an overall consistency.

New Figure 1: Mean annual heat flux components for the period of 2006-2020 computed from ECMWF (left) and ERA5 (right) daily time series

New Figure 2: Seasonal variations of heat flux components: Left column is the monthly average values for February and right column is the average for August for the period 2006-2020 (ECMWF data).

\_\_\_\_\_

Similarly related – Figure 7 shows the Qnet, but what is driving the extreme Qnet – typically I assume latent heat flux (and reduced shortwave) is the main driver of fall/winter cooling events, but Figure 1 suggests that these are quite similar between ERA5 and ECMWF at least in terms of the mean... Is that the case in the extreme events?

Thanks for your comment. Yes, we agree that latent heat flux is a major component for the fall/winter extreme events. We show now in Fig. A2 that also sensible heat flux provides considerable fall/winter extremes.

The location of the heat flux anomalies (Fig.1) is the same due to the geometry of the basin and the atmospheric forcing structure, but the extremes are different. We now show the time series of Qnet for ERA5 in Fig. A3 which in our opinion makes the point together with Fig. 8.

Figure A3: Time series of the basin averaged  $Q_{net}$  (ERA5) means,  $Q_{net}$  extremes removed, and long-term yearly climatology and four lower quantile boundary line marked with dashed lines using different k values [k=1.25, 1,0.75, 0.5].

Minor comments

Line 50-51 Awkward phasing

Thanks, sorry for the unclear phrasing. Now it is:: Moreover, the Mediterranean net heat budget comprises of several terms that show a considerable range of uncertainties (Jorda' et al., 2017).

Line 83 'have shown large deviation' - revise phrasing

Thanks, sorry for the unclear phrasing. Now it is: Using downscaled NCEP/NCAR global reanalysis of ½° x ½° resolution, Ruiz et al. (2008) computed a heat budget of -1 Wm-2. However, their heat flux components values are not close to most of the literature values (for instance, the major difference was in the value for net short wave with 84 Wm-2).

Line 162 Was  $\rho$  (rho) defined?

Added "  $\rho = 1.22 \text{ kg/m3}$ ".

Line 400 'that this is the reason why...'

Corrected and added, here we refer the Figure A3 also

Line 404-405 I don't really understand this sentence.

Thanks for suggesting an explanation, here is the new text:

The Qnet could become an impact indicator of the Mediterranean for sea level trends in the basin. The net heat budget in fact relates to the sea level tendency (Pinardi et al., 2014) in the Mediterranean Sea and could be considered as a key indicator of climate impacts in the Mediterranean Sea.

Pinardi, N., A. Bonaduce, A. Navarra, S. Dobricic, P. Oddo, 2014. The mean sea level equation and its application to the mediterranean sea. J. Climate, 27, 442–447, doi: 10.1175/JCLI-D-13-00139.1

Line 453 'Differences appear...' The main difference were in the long and short wave radiation. Figure 1 showed that latent and sensible heat fluxes were not that different.

Thanks for suggesting the clarification, the text is now: "Differences appear in the structure of the fluxes, especially the SW and LW, when different atmospheric data sets are used,..."

Line 485-487 These sentences are confusing to me. I suggest revising them in some way.

Thanks for pointing out the missing minus sign. The phrase:

"The threshold value that produces a positive basin mean heat loss is 231 W m-2. Thus, if the basin mean heat loss does not exceed this value, the basin is not in steady state. This might be a good indicator of Mediterranean Sea heat content trends to be exploited in the future."

**Is substituted by:**

"The anomaly threshold value of -231 W m-2 (Table 2) results in a long-term positive net heat flux, which is inconsistent with the basin's energy closure assumption, thereby indicating the presence of long-term changes within the basin due to atmospheric forcing"

Line 493 poor phrasing – this sentence should be revised.

Revised "Furthermore, the PDF analysis of turbulent heat fluxes will allow us to have a better understanding of the extreme events and their contributions in the net negative heat budget."

---

## Author Comment (AC3)

Reviewer comments are in black, answers from the authors are in blue and corrections added to the manuscript are in green

The paper of **Ghani et al.**, entitled 'Revisited heat budget and probability distributions of turbulent heat fluxes in the Mediterranean Sea' compared two surface heat budget estimates for the 2006-2020 period over the Mediterranean Sea recomputed by the authors with ECMWF and ERA5 bulk outputs (but the same SST). The paper presents also a statistical analyse of the sensible and latent heat fluxes with a characterization of their distributions and finally investigates the role of heat loss extremes.

12 This study has questioning conclusive remarks:

• The mention that the heat budget closure hypothesis cannot be satisfied with coarse resolution (lines 457-458) is not fully exact as shown by Table 1 where previous studies prove their quality to obtain negative heat loss in surface balanced by Gibraltar heat inflow. Possibly you would like to argue that a better representation of the heat budget is related to horizontal resolution; But there are many sources of improvements for representation of the heat budget terms: one is likely resolution, but sea surface and clouds/radiative schemes are also very important. This conclusion must be more carefully discussed in my opinion.

Thanks for your remarks. Yes, we agree that we didn't mention any other possible sources for improvement of the heat budget components. The radiative fluxes are featured with a level of uncertainty between two datasets probably due to cloud coverage, but we didn't carry further discussion in the conclusion which we will add after lines 457-458.

"In addition, it's known that the cloud coverage schemes differ between the two datasets and show a noticeable difference in the eastern Mediterranean Sea (Fig. A1), where cloud coverage distribution is denser in ECMWF data than in ERA5. This variation in cloud coverage highlights how different formulations, along with the spatial resolution of the atmospheric fields, influence cloud distribution and consequently affect the estimation of radiative flux fields."

• The statement that *the Mediterranean is still losing heat but only if using a(the) high-resolution ECMWF analysis (lines 459-460)* puzzles me. I am not sure this is a way to promote the results. The fact the Mediterranean Sea losses or not heat is something that can not be settle by only looking on one dataset. A very large analyse of a large amount of data is mandatory.

Thanks for your concern. In our study, we employed two datasets generated from one of the widely recognized atmospheric model variables applied in the Mediterranean Sea for forcing ocean forecasting models. The ERA5 reanalysis dataset, which is freely available and widely used within the scientific community, provides many atmospheric variables for model applications. However, the ECMWF operational analysis dataset is not publicly accessible,

which limits the access the availability of long time series and several key variables, such as surface heat fluxes. Importantly, as our study is connected to the context of probable uncertainties arising from the atmospheric forcing variables in ocean forecasting, these two datasets were deemed the most relevant and were therefore selected for this study.

45

- Please also clarify paragraph p10, lines 273-279. It is confusing to put here the finding is that
- 47 the Mediterranean Sea still looses heat in surface, as you decided to follow the heat budget
- 48 closure hypothesis that imposes this.
- Thanks for your concern. We move this part to conclusion and insert in modifying phase this part after the line 459, P-18.
- 51 "Our initial question was: is the Mediterranean Sea in the past 15 years still losing heat at the 52 surface? The answer is yes if we use a high-resolution ECMWF atmospheric analysis. Thus, 53 comparing the net heat flux Qnet estimates derived from ERA5 reanalysis and ECMWF 54 analysis dataset, using specific bulk formulas demonstrates an uncertainty in the results. This uncertainty, potentially linked to the spatial resolution difference between the two datasets, 55 56 impacts the regional heat budget closure. In spite of having limitation in using only two but 57 recognized atmospheric datasets, we are now able to address the question that the Mediterranean Sea in the past 15 years has still been losing heat at the surface. Therefore, 58 notwithstanding the climate change warming, the Mediterranean still looses heat due to its heat 59 budget closure constraint. However, it might be that it looses less than in the previous decades, 60 and this will be the scope of more studies in the future" 61

62

63

64

65

66 67

68

69

70

71 72

73

74 75

- I have also main concerns related to:
  - the LH distribution (section 4.2). Fig. 5a shows surprisingly a quite large number of positive LH values for all locations. This means condensation, and supersaturation of air mass. This phenomenon is quite rare. It appears mandatory to check the LH values in these distributions. Also, for turbulent fluxes, the computation uses transfer coefficients independent from the wind (equation 9/10). Does this may affect your results in terms of SH/LH distribution shapes.
- Thank you for your observation. We used the anomaly of latent heat (LH) distribution with respect to daily season cycle, where exhibits positive and negative values. We believe, there is no condensation and supersaturation dynamics present in our data, as the computed LH time series remains consistently negative. Following Petenuzzo (2010), we used constant turbulent exchange coefficients, which are multiplied by stable and unstable condition parameters and updated based on the maximum and minimum wind speeds.
- 76 There are very large differences in SW (Fig. 1d, h). This is the main reason for the ERA5
- positive budget (Tab. 1). From equation 2, I understand the differences come only from the
- 78 cloud coverage *C*. Did you compare the cloud coverage fields in the two atmospherical dataset?
- 79 Should the threshold to define clear sky be adapted and?
- 80 Thanks for your concern. Yes, we have analysed the difference in cloud coverage fields
- 81 between ECMWF analysis and ERA5 reanalysis dataset. We agree that cloud coverage
- 82 difference generated large variation in SW fields (Fig. 1 d, h). Here we present the maps of

cloud coverage (A1). Addressing your comment, we add a part that referring to the difference in cloud coverage resulting from different cloud schemes in conclusion section, which is already described under the first comment's reply above. We would prefer not to change the clear sky Reed (1977) formula because the concept is to use same formulation with different atmospheric datasets.

Figure A1: Total cloud coverage (%) mean computed for the period 2006-2020, a) ECMWF and b) ERA-5 input datasets.

For these two remarks, a larger discussion of what is mentioned p10, line 263-366 would be greatly useful.

Thanks for your feedback. But we are not sure about the line numbers, maybe it should be 263-266, because after line 291, it's another section and not related with above remarks. However, we propose to add additional text starting from line 266

"A higher spatial resolution is important for capturing many small-scale atmospheric and oceanographic features. In contrast, ERA5 comes with a horizontal resolution of ~31 km, which can smooth many small-scale features and underestimate the frequency of extreme fluxes, such as extremes that are often associated localized events – such as local wind flows, sharp air-sea temperature contrast and coastal orographic effects. With the 0.125x0.125 spatial grids, the ECMWF analysis dataset exhibits better representation of the influence of near-surface atmospheric variables, such as wind speed, air temperature and specific humidity gradient, as we notice this difference between the turbulent heat fluxes of the two datasets. In the Gulf of Lion, the local Mediterranean wind called Mistral wind influences the vaporization process significantly; a distinct gradient difference is visible in between the two datasets for latent heat fluxes (Fig.1 b & f). Even though the spatial distributions are identical in both datasets, ERA5's resolution indicates a reasonable variation in surface air-sea temperature gradient in the Alboran Sea, Adriatic Sea and Tyrrhenian Sea for turbulent heat fluxes"

Finally, even if I understand and find fair the objective of having the same fluxes computation method and same SST for both datasets, I would have appreciated a brief comparison with the SW, LW, LH and SH fluxes directly taken from ECMWF and ERA5.

We don't have the direct access to heat fluxes simulated in ECMWF analysis dataset and had requested earlier for this dataset. It would take sometimes to process and analysing at this moment, but we would be able to provide this in the revised version. Flux variables from ERA5 dataset are publicly available and presented the mean maps of those surface fluxes comparing to our computed ERA5 fluxes (Fig. A2)

Figure A2: Mean annual heat flux components for the period of 2006-2020 computed from ERA5 fluxes (left, a to d) and ERA5 inputs (right, e to h) using equations (2-10)

The basin-averaged Qnet computed using ERA5 flux variables is 4.6 Wm-2, which is approximately same to the Qnet computed using ERA5 inputs (Table 1)

127 I put below some minor comments.

p5, eq.2: add information in text about the threshold; and unit for C.

- 129 Corrected the eq.2 "if  $C \ge 0.3$  m if C < 0.3", and added in text "C (%) is the cloud coverage and converted into fraction to apply the threshold of cloud coverages ( $C \ge 0.3$ 130 and C < 0.3) adopting the formula from Reed(1977) 131 132 133 p5, line 140: what is *sec*? 134 Sec represents secant ( $sec(\theta)$ ) of zenith angel of the Sun 135 p6, eq 8: why did you not use directly the specific humidity? We followed the exact formulation used in Large (2006) and Petenuzzo (2010) using 136 dewpoint temperature data to compute specific humidity. 137 138 p7, line 189: ... the following atmospheric near-surface variables ... Corrected "the following atmospheric near-surface variables .." 139 140 Fig. 1: Could a column with difference maps for each term be added? 141 The maps are based on their original spatial resolution, and we didn't regrid the atmospheric datasets into a common grid for the comparison. 142 P8, line 210-212: ... The largest mean sensible heat gain is observed... Gulf of 143 *Lion* loss *more*... [negative is heat loss] 144 Corrected "The largest mean sensible heat gain is observed ....... Gulf of Lion loss 145 more ..." 146 p8, lines 221-223: The first reason for SW differences between Western Mediterranean and 147 Eastern Mediterranean is the latitudinal position of each sub-basin. 148 149 Added "The first reason for SW differences between Western Mediterranean and Eastern Mediterranean is the latitudinal position of each sub-basin. Furthermore, 150 radiative heat fluxes using ECMWF and ERA5 datasets are connected to different cloud 151 cover schemes. The difference in SW radiation between the western and eastern 152 Mediterranean indicates this cloud cover variation, leading to a larger heat gain in the 153 Eastern Mediterranean" 154 155 156 P8, line 227: ... presumably due to the warm Atlantic surface inflow... Thanks for your observation and corrected: "presumably due to 157 the warm Atlantic surface inflow..." 158 159 p13, line 328: From Fig. 4b μ is mostly positive. Please modify the sentence.
- Corrected "the location parameter (μ) exhibits mostly within positive value range while a small area in the Alboran Sea show negative values, ....."

- P15, line 395: ...with long term climatology values for the extreme heat losses days...: Could you precise how is built this climatology?
- Thanks for your concern and added "These extreme values were replaced with longterm daily climatological values (using equation 11) to the respective days of extremes heat losses occurred. After the replacement of extremes, the long-term basin-averaged Qnet was recalculated."
- P16, line 400-401: The differences in Qnet between ECMWF and ERA5 is mostly due to differences in SW. Please review the whole paragraph.
- Corrected and reviewed "We observed the difference in basin-averaged Qnet values 170 between ECMWF and ERA5 is primarily due to difference in SW, and the distribution 171 of extremes is likely not significantly affected by difference in spatial resolution. The 172 regions of negative heat loss, where potential extremes occur, are concentrated in the 173 174 core zones of the Gulf of Lion and Aegean Sea in both Qnet from ECMWF and ERA5 whereas a similar pattern of heat gain is evident in the Alboran Sea (Fig 3). We believe 175 that these winter extremes, featured by large negative heat loss values, significantly 176 impact the Qnet (Fig. 6), where they arise from dense heat loss area generated by local 177 Mediterranean wind flows (e.g., Mistral,) and air-sea temperature contrast. 178 Furthermore, after replacing these extremes with long-term climatology, the 179 recalculated yearly mean of seasonal climatology increased to +4 W/m, compared to -180 181 4 W/m2 when extremes were present. This confirms the influence of surface heat flux extremes in the Mediterranean net heat budget." 182
- 183 P17, line 424: ... of -289 W/m2 (for k=0.75)...
- Corrected: " -289 W/m2 (for k=0.75)"
- p17, line 427: Please add the map of differences between Fig8 and Fig. 3a to put in evidencethis result.
- Yes, we have added this new figure replacing Fig. 8

Figure 8: The annual mean  $(Q_{net})$  after the removal of extremes showing significant reduction of negative heat fluxes in the Gulf of Lion, Adriatic Sea and Aegean Sea regions.

P19, line 486: minus sign is missing.

194 - Corrected : " -238 W/m2"

---

## Author Comment (AC4)

Reviewer comments are in black, answers from the authors are in blue and corrections added to the manuscript are in green Anonymous reviewer 2 The paper of Ghani et al., entitled 'Revisited heat budget and probability distributions of turbulent heat fluxes in the Mediterranean Sea' compared two surface heat budget estimates for the 2006-2020 period over the Mediterranean Sea recomputed by the authors with ECMWF and ERA5 bulk outputs (but the same SST). The paper presents also a statistical analyse of the sensible and latent heat fluxes with a characterization of their distributions and finally investigates the role of heat loss extremes.

10 -----

11 This study has questioning conclusive remarks:

The mention that the heat budget closure hypothesis cannot be satisfied with coarse resolution (lines 457-458) is not fully exact as shown by Table 1 where previous studies prove their quality to obtain negative heat loss in surface balanced by Gibraltar heat inflow. Possibly you would like to argue that a better representation of the heat budget is related to horizontal resolution; But there are many sources of improvements for representation of the heat budget terms: one is likely resolution, but sea surface and clouds/radiative schemes are also very important. This conclusion must be more carefully discussed in my opinion.

We agree that spatial resolution may not be the only factor causing the difference. Our method allows to eliminate the SST as a possible cause, as noted by the reviewer. We agree with the reviewer that the distribution of cloud cover is also an important difference (reported in Fig A1 below). In fact, the difference is a complex function of different quality of the atmospheric variables. To be noted is that ERA5 and ECMWF analyses use approximately the same model and data assimilation systems. However, we have added also the consideration of cloud cover among the potential differences.

We would like to point out that we had already a sentence at line 223: "Furthermore, ECMWF and ERA5 different values are connected to different cloud cover."

Figure A1: Total cloud coverage (%) mean computed for the period 2006-2020, a) ECMWF and b) ERA-5 input datasets.

To make a more balanced statement we have eliminated in the abstract the sentence at line 35 "This highlights the importance of high-resolution atmospheric data for accurately capturing air-sea interactions and ensuring physically consistent climate modelling over the Mediterranean Sea."

replacing it with:

"Only ECMWF fields are consistent with the heat budget closure hypothesis."

40 -----

The statement that the Mediterranean is still losing heat but only if using a(the) high-resolution ECMWF analysis (lines 459-460) puzzles me. I am not sure this is a way to promote the results. The fact the Mediterranean Sea losses or not heat is something that cannot be settle by only looking on one dataset. A very large analyse of a large amount of data is mandatory.

Thanks for your concern. Our study is a conceptual study of how two different data sets give different heat balances using the same air-sea flux formulations and SST. The ECMWF analyses and reanalyses are among the most widely used data sets for the Mediterranean Sea and they are special since they assimilate available observations. Yet no specific study is found in the literature. As listed in Table 1, almost all the previous studies have been done with single datasets but not with analyses and reanalyses. We thank the reviewer for forcing us to specify this important novelty of our paper.

We have now modified the introduction at line 60:

55 Furthermore, the estimate of the Mediterranean Sea heat budget from ECMWF meteorological analysis data sets has not been done before. 56 After line 76 we discuss the hypothesis behind the "closure of the heat budget": 57 58 We realise that assuming perfect balance between lateral and vertical heat fluxes, even in the Mediterranean Sea, is an approximation. Being heat clearly entering the Mediterranean Sea 59 through Gibraltar, we search for a negative net heat flux, which we call the closure hypothesis. 60 How negative such net heat flux is, we do not know but searching for a negative value is a 61 conservative assumption aligned with current scientific understanding. 62 63 64 Please also clarify paragraph p10, lines 273-279. It is confusing to put here the finding is that the Mediterranean Sea still loses heat in surface, as you decided to follow the heat budget 65 closure hypothesis that imposes this. 66 67 Thanks for your concern. We do not impose the negative heat budget; we check if the data sets can give a negative net heat flux using the same bulk formulas and SST for ECMWF and ERA5 68 69 surface fields. We have moved all the lines 273-279 to the conclusion section inserting a modified phrase 70 71 after the line 459, P-18 72 "Our initial question was: is the Mediterranean Sea in the past 15 years still losing heat at the surface? The answer is yes if we use ECMWF atmospheric analyses. Additionally, comparing 73 74 the *Q*net estimates derived from ERA5 and ECMWF with the same bulk formulas demonstrates that the uncertainty peaks in the atmospheric forcing resolution and possibly cloud cover. This 75 76 uncertainty impacts the regional heat budget closure hypothesis." 77 78 79 I have also main concerns related to: the LH distribution (section 4.2). Fig. 5a shows surprisingly a quite large number of positive 80 81 LH values for all locations. This means condensation, and supersaturation of air mass. This phenomenon is quite rare. It appears mandatory to check the LH values in 82 these distributions. Also, for turbulent fluxes, the computation uses transfer coefficients 83 84 independent from the wind (equation 9/10). Does this may affect your results in terms of SH/LH distribution shapes. 85 86 Thanks for asking this question. We used the anomaly of latent heat (LH) distribution with respect to daily season cycle, where exhibits positive and negative values. We believe, there is 87 no condensation and supersaturation dynamics in the full computed LH time series which 88 remains consistently negative. Following Pettenuzzo et al. (2010), we used constant turbulent 89 exchange coefficients, which are multiplied by stable and unstable condition parameters and 90 updated in final computation using maximum and minimum wind speeds condition. 91 92 93

- 94 There are very large differences in SW (Fig. 1d,h). This is the main reason for the ERA5
- 95 positive budget (Tab. 1). From equation 2, I understand the differences come only from the
- 96 cloud coverage C. Did you compare the cloud coverage fields in the two atmospherical
- 97 dataset? Should the threshold to define clear sky be adapted and?
- We agree that cloud coverage difference generated large variation in SW fields (Fig. 1 d, h).
- 99 Here we presented the maps of cloud coverage (Fig A1). We would prefer not to change the
- 100 clear sky Reed (1977) formula because the concept is to use same formulation with different
- 101 atmospheric datasets.
- For these two remarks, a larger discussion of what is mentioned p10, line 263-366 would be
- 103 greatly useful.
- Thanks for your comment. We think we have already clarified this part in the answer to your
- initial remark.
- 106 -----
- Finally, even if I understand and find fair the objective of having the same fluxes computation
- method and same SST for both datasets, I would have appreciated a brief comparison with the
- 109 SW, LW, LH and SH fluxes directly taken from ECMWF and ERA5.
- 110 ECMWF analysis datasets do not provide directly surface fluxes but only forecast fluxes,
- initialized from the analyses. However, ERA5 contains fluxes, and we have now plotted them
- in Fig. A2 below.
- First, we point out that the net heat budget from the ERA5 fluxes is +5.3 W/m2 (Fig A2, left
- panel), which is again positive. Secondly, we see that the major difference is in the LW and
- SW radiative components, but the changes compensate giving a similar net radiative balance
- 116 (Fig A3)

Figure A2: Mean annual heat flux components for the period of 2006-2020 computed from ERA5 fluxes (left) and ERA5 inputs (right) using equations (2-10)

Figure A3: Differences (ERA5 fluxes - Computed fluxes) of the ERA5 fluxes and computed fluxes using ERA5 atmospheric inputs for LW and SW

126 I put below some minor comments.

p5, eq.2: add information in text about the threshold; and unit for C.

- Corrected the eq.2 "if  $C \ge 0.3$  and if C < 0.3", and added in text "C (%) is the cloud coverage converted into fraction.

p5, line 140: what is sec?

- Sec represents secant ( $sec(\theta)$ ) of zenith angel of the Sun

- p6, eq 8: why did you not use directly the specific humidity?
- We followed the exact formulation used in Large (2006) and Petenuzzo (2010) using dewpoint temperature data to compute specific humidity.
- p7, line 189: ... the following atmospheric near-surface variables...
- Corrected "the following atmospheric near-surface variables"
- 138 Fig. 1: Could a column with difference maps for each term be added?
- The maps are based on their original spatial resolution, and we didn't regrid the atmospheric datasets into a common grid for comparison.
- 141 P8, line 210-212: ... The largest mean sensible heat gain is observed... Gulf of
- 142 *Lion* loss *more*... [negative is heat loss]
- Corrected "The smallest mean sensible heat loss is observed ....... Gulf of Lion loses more .."
- p8, lines 221-223: The first reason for SW differences between Western Mediterranean and
- Eastern Mediterranean is the latitudinal position of each sub-basin.
- 147 Added "The first reason for SW differences between Western Mediterranean and Eastern
- Mediterranean is the latitudinal position of each sub-basin. Furthermore, radiative heat fluxes
- 149 using ECMWF and ERA5 datasets are connected to different cloud cover schemes. The
- difference in SW radiation between the western and eastern Mediterranean indicates the cloud
- cover differences, leading to a larger heat gain in the Eastern Mediterranean"
- 152 Do 1:00
- 153 P8, line 227: ... presumably due to the warm Atlantic surface inflow...
- Corrected: "presumably due to the warm Atlantic surface inflow..."
- 155 p13, line 328: From Fig. 4b μ is mostly positive. Please modify the sentence.
- Corrected "the location parameter (μ) exhibits mostly positive values while a small area in the Alboran Sea show negative values, ....."
- P15, line 395: ...with long term climatology values for the extreme heat losses days...: Could you precise how is built this climatology?
- We added: "These extreme values were replaced with long-term daily climatological values (using equation 11) to the respective days of extremes heat losses occurred"
- P16, line 400-401: The differences in Qnet between ECMWF and ERA5 is mostly due to
- differences in SW. Please review the whole paragraph.
- We thank the reviewer for pointing out this unclear statement. In this picture we discuss the
- sensitivity of long term Qnet basin average values to the extremes of the time series shown in

Fig. 6. In Table 2 we show that the Qnet without extremes becomes positive, as it is for ERA5 in Table 1. It is true that a comparison between extremes of ERA5 and ECMWF has not been done for the extreme, but it will be somewhat irrelevant since ERA5 has already a Qnet positive.

We have substituted the phrase at lines 400-401 with the following:

We argue that the ECMWF net heat extremes are the reason why ECMWF has a negative long term mean budget.

172 P17, line 424: ... of -289 W/m2 (for k=0.75)...

**Corrected thanks**

166

167

168

169

170171

173

174

175

p17, line 427: Please add the map of differences between Fig8 and Fig. 3a to put in evidence this result.

New Figure 8: The annual mean after the removal of extremes with significant reduction of negative heat fluxes in the Gulf of Lion, Adriatic Sea and Aegean Sea regions.

P19, line 486: minus sign is missing.

Corrected, thanks

According to my main remarks, I recommend a major revision of the paper.

182

176

177178

179180

181